# SYMMETRICAL SYNCMAP FOR IMBALANCED GENERAL CHUNKING PROBLEMS

## ABSTRACT

Recently, SyncMap (2021) pioneered an approach to learn complex structures from sequences as well as adapt to any changes in underlying structures. Such approach, inspired by neuron group behaviors, is achieved by using self-organizing dynamical equations without any loss functions. Here we propose Symmetrical SyncMap that goes beyond the original work to show how to create dynamical equations and attractor-repeller points which are stable over the long run, even dealing with imbalanced continual general chunking problems (CGCPs). The main idea is to apply equal updates from positive and negative feedback loops by *symmetrical activation*. We then introduce the concept of *memory window* to allow for more positive updates. Our algorithm surpasses or ties other unsupervised state-of-the-art baselines in all 12 imbalanced CGCPs with various difficulties, including dynamical ones. To verify its performance in real-world scenarios, we conduct experiments on several well-studied structure learning problems. The proposed method surpasses substantially other methods in all scenarios, suggesting that symmetrical activation plays a critical role in uncovering topological structures and even hierarchies encoded in temporal data.

## 1 INTRODUCTION

Human brains have been proved to have unsupervised abilities to detect repetitive patterns in sequences involving texts, sounds and images (Orbán et al., 2008; Bulf et al., 2011; Strauss et al., 2015). In the field of neuroscience, part of this behavior is known as chunking. Chunking has been verified in many experiments to play an important role in a diverse range of cognitive functions (Schapiro et al., 2013; Yokoi & Diedrichsen, 2019; Asabuki & Fukai, 2020). Related to chunking problems, many sequence processing algorithms in machine learning have been proposed for time-series clustering (Aghabozorgi et al., 2015) based on similarity measurements (Figure 1(a)). Chunking sequences between state variables, however, is still underexplored (see Figure 1(b)(c)).

Recently, Vargas & Asabuki (2021) proposed the first learning of chunking based solely on self-organization called SyncMap. The authors also extended chunking problems into one called Continual General Chunking Problem (CGCP), which includes problems with diverse structures that can change dynamically throughout the experiments. For the first time, SyncMap was shown not only able to uncover complex structures from sequential data, but also to adapt to continuously changing structures. It achieves this with self-organizing dynamics that maps temporal input correlations to spacial correlations, where the dynamics are always updating with negative/positive feedback loops. In this work, however, we identify problems in the original dynamics that lead to long-term instability, and we further show that performances in imbalanced CGCPs are poor given the asymmetric number of updates, i.e., the number of negative updates is much bigger than that of the positive ones.

Beyond identifying these problems, here we propose Symmetrical SyncMap, which can solve both of the problems above using symmetric selection of nodes and generalized memory window. Symmetrical SyncMap solves the instability of the dynamics efficiently, and goes beyond to propose a solution to deal with imbalanced general chunking problems. As opposed to the original SyncMap that suffers from the uneven updates from positive/negative feedback loops, we propose *symmetrical activation*, and further introduce the concept of *memory window*, so that the system can have more updates from positive feedback loop while concurrently reducing the number of negative updates. In fact, the symmetrical number of updates not only compensates when imbalanced chunks are

presented, but also makes the algorithm stable over the long run and reaches an equilibrium quickly in changing environments. By showing that equilibrium and self-organization can appear only with dynamical equations and without optimization/loss functions, the biggest motivation from this paper is realizing how the substantial improvements, beyond the self-organization inspiration, make the new learning paradigm very adaptive and precise. Moreover, the simplicity of the modifications here, as supported by the effectiveness in real-world scenarios of structure learning, solves the problem at the foundation, while keeping the final method concise and improving it in both accuracy and stability.

## 2 RELATED WORKS

**Chunking.** Natural neural systems are well known for the unsupervised adaptivity, since they can self-organize by many mechanisms for several purposes on many timescales (Lukoševicius, 2012). One of the mechanisms is chunking, which can be described as a biological process where the brain attains compact representation of sequences (Estes et al., 2007; Ramkumar et al., 2016). Specifically, long and complex sequences are first segmented into short and simple ones, while frequently repeated segments are concatenated into single units (Asabuki & Fukai, 2020). This can be seen as a complexity reduction for temporal information processing and associated cost (Ramkumar et al., 2016).

Albeit our focus is more on neuroscience and machine learning perspectives, earlier algorithms proposed for solving chunking problems are from linguistics and include PARSER (Perruchet & Vinter, 1998). It performs well in detecting simple chunks, but fails when the probability of state transition are uniform (Schapiro et al., 2013). A neuro-inspired sequence learning model, Minimization of Regularized Information Loss (MRIL) was proposed by applying a family of competitive network of two compartment neuron models that aims to predict its own output in a type of self-supervised neuron (Asabuki & Fukai, 2020). Albeit the interesting paradigm, MRIL has been shown unstable even for problems in which it performs reasonably well. Very recently, a self-organizing learning paradigm (called SyncMap) has been proposed, which surpassed MRIL in all scenarios (Vargas & Asabuki, 2021).

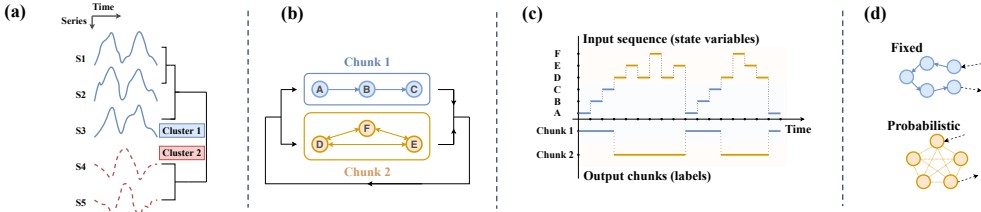

Figure 1: **Explanation of the difference between time-series clustering and sequence chunking.** (a) Process of time-series clustering. Homogenous time-series (S1-S3, S4-S5) are grouped together based on a certain similarity measure. (b) Chunking problems example. A fixed chunk (state variables A-B-C [blue]) and a probabilistic chunk (D-E-F [orange]) are repeated in the input sequence with equal probabilities. (c) Input-output map of problem structure in (b) over time. State transitioning by first-order Markov chain. (d) Examples of the structures of fixed chunk and probabilistic chunk.

**Time-series Clustering.** Time series data is defined as a sequence of continuous, real-valued elements, usually with high dimensionality and large data size (Aghabozorgi et al., 2015). As a subroutine in unsupervised sequence processing, time-series clustering aims to uncover patterns, usually in very large sequential datasets that cannot be manually handled. This can be found in some articles applying competition-based self-organizing maps (SOMs) (Kohonen, 1990) and their variations (Vannucci & Colla, 2018; Fortuin et al., 2018), which are well-suited for clustering time series but not capable of chunking time series. In other words, these SOMs were not designed to find the underlying structures of sequences and correlation between variables, therefore, their objectives are different. A comparison of time-series clustering and sequence chunking is shown in Figure 1.

**Word Embeddings.** In the field of natural language processing, word embedding algorithms generally transforms texts and paragraphs into vector representations (Khattak et al., 2019; Bojanowski et al., 2017; Peters et al., 2018). FastText enriched the word vector with subword information (Bojanowski et al., 2017), whereas ELMo (Peters et al., 2018) and BERT (Devlin et al., 2018) aimed to represent word by contextualized word embeddings. Chunking problems presented here are related to some

of them, such as a prediction-based Word2vec embedding algorithm (Mikolov et al., 2013) that transforms texts into a vector space representation and can be combined with clustering to deal with chunking problems. Therefore, Word2vec is used in the experiments.

**Representation Learning and Communities Detection.** The problem of finding probabilistic chunks refers to a random walk over a graph with several chunk structures; in which the possibility of transition to an internal state within a chunk is higher than that of transition to an external state belonging to other chunks. Such graph structures mentioned above can be seen as communities (Radicchi et al., 2004), which are most-studied by recent representation learning algorithms such as DeepWalk (Perozzi et al., 2014) and Graph Neural Networks (GNNs) (Kipf & Welling, 2016). More related, the Modularity Maximization (Newman & Girvan, 2004; Tang & Liu, 2009) uses eigen-decomposition performed on the modularity matrix to learn vertex representation of community. By using the adjacency matrix (transition probability matrix) to convert sequential data to graph structure, Modularity Maximization can also deal with chunking problems via random walk over the generated graphs. Although there exists newer modularity-based algorithms which try to optimize the pioneering work, such as Louvain method (Blondel et al., 2008), their objective is mostly to reduce the computational cost. Therefore, we use the original Modularity Maximization in the comparison.

## 3 CONTINUAL GENERAL CHUNKING PROBLEMS

Recently, a problem called Continual General Chunking Problem (CGCP) has been first proposed by Vargas & Asabuki (2021). The paper generalized various problems from neuroscience to computer science, including chunking, causal and temporal communities and unsupervised feature learning of time sequences. Such problems are considered as extracting co-occurring states from time sequences, in which the generation process (i.e., data structure) can change over time. To illustrate, the input sequences of CGCP contain state variables where each state belongs to a fixed chunk or a probabilistic chunk, transitioning by first-order Markov chain. The element of the transition matrix is given by: $P_{ab} = Pr[\boldsymbol{s}_{t+1} = \boldsymbol{b} | \boldsymbol{s}_t = \boldsymbol{a}]$, where $\boldsymbol{s}_t$ is the state vector at $t$; and $\boldsymbol{a}$ and $\boldsymbol{b}$ are the label of states.

Specifically, the fixed chunk problem (see Figure 1(d) the blue chunk) refers to the situation that the next state $\boldsymbol{s}_{t+1}$, with respect to the current state $\boldsymbol{s}_t$, is deterministic within a chunk. For example, if $\boldsymbol{a}$ and $\boldsymbol{b}$ are two continuous elements of a fixed chunk with direction $\boldsymbol{a}$ to $\boldsymbol{b}$, then $P_{ab} = 1, \sum_b P_{ab} = 1$.

More realistically, the probabilistic chunk problem (see Figure 1(d) the orange chunk) refers to input sequences which are generated by giving a random walk over graphs. The graphs are characterized by two types of degrees: internal degree $k_i^{int}$ and external degrees $k_i^{ext}$ of state $i$. For every state, the following constrain holds: $k_i^{int} > k_i^{ext}$ for all $i \in A$, where $A$ is the set of all states in sequences. The above constraint is satisfied if the graph has dense connections within chunks but sparse connections between nodes in different chunks.

Beyond generalizing chunking problems to fixed and probabilistic chunks, CGCP also considers their continual variations. This is motivated by the constant adaptation observed by neural cells that can relatively switch behavior quickly in different environments (Dahmen et al., 2010). In this case, the data structure can change over time, conferring a harder albeit realistic setting.

## 4 SYNCMAP

SyncMap is a sole self-organization based algorithm proposed by Vargas & Asabuki (2021). It solves CGCP by creating a projected map that encodes the temporal correlation (chunks) as spatial distance between nodes. In SyncMap's dynamic, nodes which are activated together tend to be grouped as chunks, while nodes that do not activate together will be pulled away from each other. The algorithm is explained in detail in the following, with a brief example shown in Figure 2(a).

**Input Encoding.** Consider an input sequence of state variables $\boldsymbol{S} = \{\boldsymbol{s}_1, \boldsymbol{s}_2, ..., \boldsymbol{s}_t, ..., \boldsymbol{s}_\tau\}$, where $\tau$ is the sequence length. $\boldsymbol{s}_t = \{s_{1,t}, ..., s_{n,t}\}^T$ is a vector at time step $t$, and its elements $s_{i,t}$, $i = 1, ..., n$ hold constrain $s_{i,t} \in \{0, 1\} : \sum_{i=1}^{n} s_{i,t} = 1$, where $n$ is the number of states. The input is encoded as an exponentially decaying vector $\boldsymbol{x}_t = \{x_{1,t}, ..., x_{n,t}\}^T$ having the same shape as $\boldsymbol{s}_t$:

$$x_{i,t} = \begin{cases} s_{i,t_a} * e^{-0.1*(t-t_a)}, & t - t_a < m * tstep \\ 0, & otherwise \end{cases} \tag{1}$$

where $t_a$ is the last state transition to state $\boldsymbol{s_i}$, and $m$ is the state memory. Specifically, state transitions happen every $tstep$ step (also known as time delay), and variables that have their activation period greater than $m * tstep$ are set to 0. $m$ and $tstep$ are set at 2 and 10 in the original work. An example of the encoded exponentially decaying input is shown in Figure 2(a).

**Training Dynamic.** We generate weight nodes $w_{i,t}$ in SyncMap's map space to obtain pair tuple $(x_{i,t}, w_{i,t})$. Nodes are first randomly initialized in a $k$ dimensional map space. Note that weight node $w_{i,t} \in \mathbb{R}^k$ is a point in SyncMap's map space, and it can also be considered as a vector.

In every iteration when a new input vector $\boldsymbol{x}_t$ comes in, all its elements $x_{i,t}$, together with the corresponding nodes $w_{i,t}$, are divided into two sets according to the threshold value $a$: (1) activated or recently activated (positive) set $PS_t = \{i|x_{i,t} > a\}$ and (2) non-recently activated (negative) set $NS_t = \{i|x_{i,t} \leq a\}$. The original SyncMap used $a$ directly at 0.1. In this paper we introduce threshold value $a$ which allows us to achieve more general state memory implementation.

Inside the space, the centroids of $PS_t$ and $NS_t$ sets are calculated as follows if and only if the cardinality of both sets are greater than one in this iteration (i.e., $|PS_t| > 1$ and $|NS_t| > 1$):

$$cp_t = \frac{\sum_{i \in PS_t} w_{i,t}}{|PS_t|}, \quad cn_t = \frac{\sum_{i \in NS_t} w_{i,t}}{|NS_t|} \tag{2}$$

where $cp_t$ and $cn_t$ are the centroids of $PS_t$ and $NS_t$ respectively. Finally, node $w_{i,t}$ corresponding to each input $x_{i,t}$ is updated:

$$\phi_{i,t} = \begin{cases} 1, & i \in PS_t \\ 0, & i \in NS_t \end{cases}, \alpha = \begin{cases} \alpha, & i \in PS_t \cup NS_t \\ 0, & otherwise \end{cases} \tag{3}$$

$$w_{i,t+1} = w_{i,t} + \alpha \left( \frac{\phi_{i,t}(cp_t - w_{i,t})}{||w_{i,t} - cp_t||} - \frac{(1 - \phi_{i,t})(cn_t - w_{i,t})}{||w_{i,t} - cn_t||} \right) \tag{4}$$

where $\alpha$ is the learning rate and $||\cdot||$ is the Euclidean distance. Subsequently, updated nodes are normalized to be within a hyper-sphere having radius of 10 at the end of the iteration.

**Clustering Phase.** SyncMap forms a map during dynamic training, which has the number of nodes equal to the number of input states $n$. After training, DBSCAN is used (Schubert et al., 2017) for clustering with the pre-defined density parameters $eps$ and minimum cluster $mc$, as it does not require the number of clusters as input.

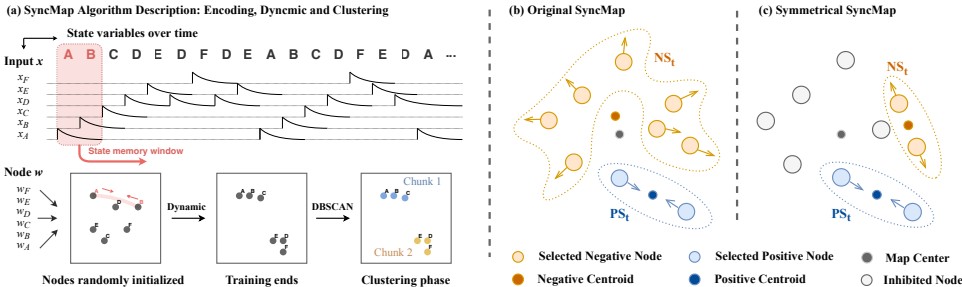

Figure 2: **Dynamics in SyncMap.** (a) Chunking procedures using SyncMap. Sequences with state variables are first encoded as exponentially decaying input $\boldsymbol{x}_t$ (top). Here, the state memory $m$=2. Weight nodes $w_i$ in SyncMap's dynamic are randomly initialized (bottom left). The dynamic is then trained by Equations 2 to 4 (bottom middle). Finally, in the clustering phase, DBSCAN is applied to obtain chunks/communities (bottom right). (b) Illustration of the instability in SyncMap. In the original work, the dynamical equations are strongly influenced by negative nodes, since the cardinality of all non-activating nodes $NS_t$ are usually much greater than that of activating nodes $PS_t$ (e.g., 7 : 2). (c) The proposed symmetrical activation. By applying stochastic selection, equal number of positive and negative nodes are activated in each iteration (e.g., 2 : 2).

**Limitations of SyncMap.** Although SyncMap shows capabilities to address all kinds of CGCP, one crucial issue is the instability of its dynamic in the long term. This is due to the asymmetric number of updates with respect to positive and negative nodes. Figure 2(b) shows how this happens with an example of nine nodes in 2-D SyncMap. The fixed state memory ($m = 2$) results in an uneven update of positive (2) and negative (7) nodes, i.e., the dynamic's update is more influenced by negative feedback loop, which causes an undesirable convergence in the long run.

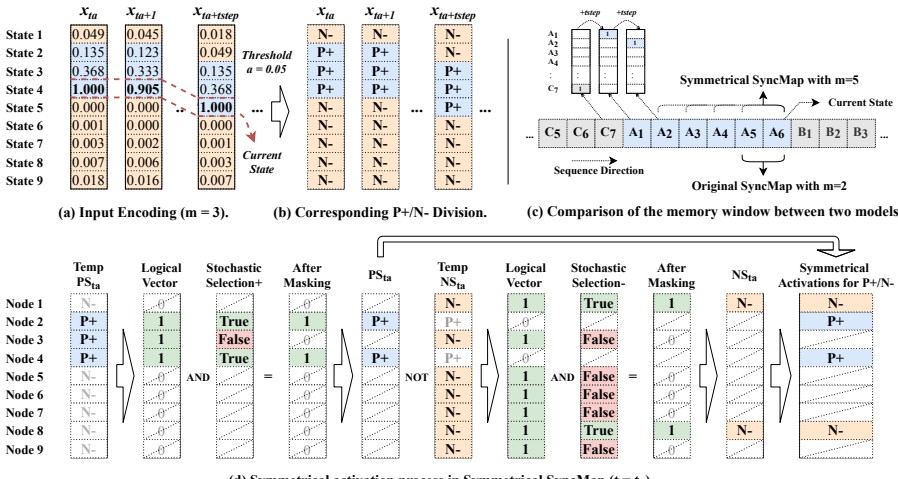

(a) Input Encoding (m = 3).  (b) Corresponding P+/N- Division.  (c) Comparison of the memory window between two models.

(d) Symmetrical activation process in Symmetrical SyncMap (t = $t_a$).

Figure 3: **General workflow of symmetrical activation.** (a) Exponentially decaying input sequence. (b) Sequence after the process of positive and negative nodes' division (represented as logical vectors). (c) Comparison of the memory window between the original and Symmetrical SyncMap. State $A_i$, $B_i$ and $C_i$ belong to three fixed chunks. The original SyncMap deterministically activates $m$ most recent states (i.e., $A_5$ and $A_6$), while Symmetrical SyncMap generalizes the state memory to have a larger memory window (e.g., from $A_2$ to $A_6$ when $m$=5) for stochastically selecting and activating positive nodes. (d) Process of stochastic selection to achieve symmetrical activation. First, we randomly lose sight of some nodes in $PS_t$ set (i.e., instead of activating all positive nodes, we stochastically select some of them to activate). This is achieved by an AND operation to the input and a masking vector having random logical values at each time step. Next, for stochastic selection of the negative nodes in $NS_t$ set, we use a masking vector similarly to that in positive part, and end up with activating equal number of positive and negative nodes in every iteration.

## 5 SYMMETRICAL SYNCMAP

Inspired by how *neural efficiency* influences brain activation by focusing the energy on smaller brain areas (Neubauer & Fink, 2009); here we propose an algorithm called Symmetrical SyncMap to better solve CGCP, particularly the imbalanced chunking problems. The main idea is to use symmetrical positive and negative activations. In other words, we try to reduce the number of activated negative nodes while selecting and activating more positive nodes in every iteration, thus balancing the updating rates in negative and positive feedback loops. To achieve symmetrical activation, we introduce *stochastic selection* and *memory window*.

### 5.1 MEMORY WINDOW: GENERALIZING THE STATE MEMORY

We introduce memory window to our algorithm by generalizing the state memory $m$, which allows a wider window for updates from the positive feedback loop, thus helping to capture the bigger chunks. This is achieved by tuning the threshold value $a$ as mentioned in SyncMap's definition, i.e., any nodes $w_{i,t}$ having its corresponding input value $x_{i,t}$ greater than $a$ will be divided into $PS_t$ set and vice versa (a true or false logical operation). With a pre-defined $tstep$, one can easily adjust threshold $a$ to control the state memory $m$ ($tstep$=10, $a$=0.05 and $m$=3 in Figure 3(a)).

### 5.2 SYMMETRICAL ACTIVATION

Symmetrical activation is the core of our proposed algorithm, where equal number of positive and negative nodes are selected to activate at each time step. We propose stochastic selection to select nodes without bias in $PS_t$ and $NS_t$ sets. Details are shown in Figure 3.

**Stochastically select nodes into $PS_t$ set.** With a pre-defined state memory $m$, we first obtain the temporary $PS_{temp}$ set in a same way the original SyncMap obtains $PS_t$ (i.e., $PS_{temp}$ includes $m$

positive nodes, and $PS_{temp} \subseteq W_t$, where $W_t = \{w_{i,t}|i = 1, ..., n\}$ is the set including all nodes). Then, we apply stochastic selection to select positive nodes into $PS_t$ (i.e., a sampling process). Whether to enable stochastic selection at this particular time step is determined by a probability parameter $Pr \in [0, 1]$. In other word, if stochastic selection were enabled, we randomly select 2 positive nodes and "inhibit" (ignore) other $m - 2$ nodes in $PS_{temp}$, with the probability of $Pr$ when state memory $m > 2$; otherwise we select all $m$ nodes. When $m = 2$, stochastic selection is not used and two most-recent states are selected. Afterwards, $PS_t$ is updated, which only includes those activated positive nodes ($PS_t \subseteq PS_{temp}$). Additionally, we give an analysis of $Pr$ in Appendix F.

**Stochastically select nodes into $NS_t$ set.** After obtaining the above $PS_t$, we define the temporary negative set $NS_{temp} = W_t - PS_t$. One may notice that there is a chance that some nodes in $PS_{temp}$ could potentially be sampled as nodes in $NS_{temp}$. This is desirable as it introduces a more uniform selection process and produces more robust results. Next, we again use the stochastic selection for sampling several negative nodes in $NS_{temp}$ set. The number of negative nodes being selected is symmetrically equal to the cardinality of $PS_t$ (i.e., $|NS_t| = |PS_t|$). After this step, the $NS_t$ set is updated ($NS_t \subseteq NS_{temp}$, see the right part of Figure 3(d) for example).

The remaining steps follow the Equations 2-4. We calculate a moving average of 10000 steps of nodes' position and use it for DBSCAN, instead of applying DBSCAN to the map at a "snapshot" time step in the original work. Algorithmic description is shown in Appendix G.

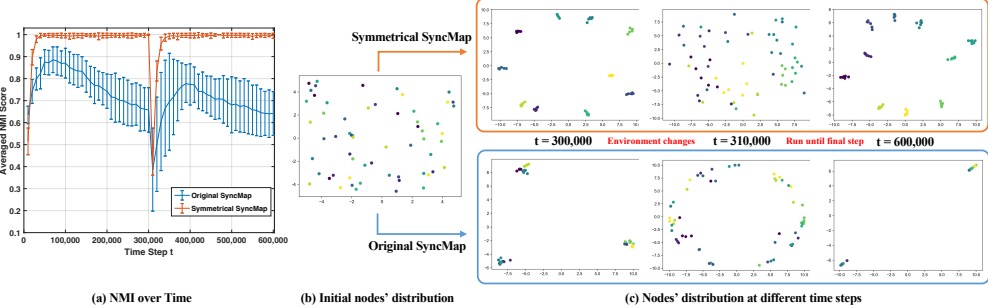

(a) NMI over Time  (b) Initial nodes' distribution  (c) Nodes' distribution at different time steps

Figure 4: **Long term behavior analysis of the original SyncMap and Symmetrical SyncMap.** (a) NMI over time of two models. Data are mean with s.t.d (error bar). Statistics are in Appendix H. (b) For comparison, we show a single trial performed by two models, initialized identically. (c) Nodes' distribution at different time steps. Colors of the nodes indicate the true labels of chunks.

## 6 EXPERIMENTS OF CGCP PROBLEMS

We evaluate the proposed Symmetrical SyncMap with 13 tests, including (i) a long-term behavior analysis, and (ii) 12 imbalanced CGCP with fixed, probabilistic and mixed chunks, as well as their continual variations. Among the large number of clustering quality measurements, we used Normalized Mutual Information (NMI) (Studholme et al., 1999) for measurement (see Appendix A for math detail). NMI ranges between 0 and 1, and the higher the score, the better the chunking performance. We did 30 trials for each experiment. Results of NMI are shown in Figure 4 and Table 1. We used a t-test with p-value of 0.05 to verify if the best result is statistically significantly different from the others (statistical results are in Appendix H). An ablation study and computation time analysis are also investigated in Appendices I and J.

### 6.1 LONG TERM BEHAVIOR ANALYSIS EXPERIMENT

To evaluate the behavior of the algorithm over the long run, we set up this experiment. The problem considered here is a continual changing environment with ten fixed chunks, each containing six different states. Transitions between chunks happen at the end of a chunk sequence (i.e., after the sixth state variable presented inside a chunk). A chunk can transit to any other chunk with equal probability. Sequence length was set to $\tau = 600000$. At time step $t = 0$ the first environment was initialized. After $t = 300000$ the problem changed to the second environment by re-assigning all 60 states into new ten chunks. We applied the original SyncMap and the proposed one in this experiment.

We used the same parameter settings of the two models, where $\alpha$=0.001*$n$, $k$=2, $eps$=1 and $mc$=2. Besides, $m$=3 and $Pr$=30% were used for Symmetrical SyncMap training.

Results in Figure 4 show that Symmetrical SyncMap reaches near the optimal performance in this experiment. By applying symmetrical activation, Symmetrical SyncMap can have long-term stability while keeping NMI near 1.0. In contrast, the NMI of the original SyncMap reaches the peak near 0.88 at $t$=70000 and decreases constantly afterwards. After environment changes, Symmetrical SyncMap detects new chunks and reaches to a new equilibrium quickly, while the original SyncMap performs poorer and becomes unstable in the long run.

## 6.2 IMBALANCED CGCP PROBLEMS

**Baselines and Parameter Settings.** We test several imbalanced CGCP problems and their continual variations by using *Symmetrical SyncMap, SyncMap, Modularity Maximization (Modularity Max), Word2vec and MRIL*. In detail, Symmetrical SyncMap's parameters were set to $\alpha$=0.001*$n$, $k$=3, $m$=3, $Pr$=30%, $eps$=4.5 and $mc$=2. We conducted a parameter sensitivity analysis shown in Appendix F. For the original SyncMap, we used $k$=3, $m$=2, $eps$=4.5 and $mc$=2. Regarding the Modularity Max, we first converted the input sequence to transition probability (TP) matrix, and then used the TP matrix to generate a graph for communities detection. To evaluate how a word embedding algorithm would fair in CGCP, a skip-gram Word2vec algorithm was modified to suit in the context of CGCP. Here, a latent dimension of 3 and an output layer with softmax were used, and the output size is equal to the inputs. Learning rate was set at 0.001 and batch size was 64 with a mean squared error as loss. A window of 100 steps (equivalent to 10 state transitions) was used to compute the output probability of skip-gram. Regarding the MRIL, we used 5 output neurons for all experiments, with the learning rate of 0.001. We gathered the output neurons showing correlation larger than 0.5, detecting chunks by assigning an index of groups that maximally respond to each input. The input sequences of all baselines were the same exponential decaying input as used in Symmetrical SyncMap.

**Problem Settings (Appendix B).** We first consider several environments which consist of 3 different sizes of chunks: big, moderate and small chunks. Specifically, the big chunk has 20 state variables, while the moderate and small chunks have 10 and 5 respectively. Based on the chunk settings, we then designed three types of imbalanced problems: (i) Two big and one small chunks (20-20-5). (ii) One big, one moderate and one small chunks (20-10-5). (iii) One big and two small chunks (20-5-5). We tested these three types of imbalanced problems with three different structure settings: fixed, probabilistic and mixed chunks tests. The structures of the fixed and probabilistic environments are shown in Figure 1(d). Please refer to Figure 6 in Appendix B for the examples of the complete structures. Regarding the mixed tests, two probabilistic chunks and one fixed chunk were presented in each environment, where the order of chunks in the input sequence was specified as: $1^{st}$ probabilistic to fixed to $2^{nd}$ probabilistic chunk. Sequence length $\tau$ is set at 200000 for all types of test.

**Dynamical Continual Variation.** Three dynamical variations of the above-mentioned problems were presented: continual fixed, continual probabilistic and continual mixed. Sequence length was set to $2\tau$. States were permuted between chunks: at time step $t$=0 the first type of problem was 15-15-5 (see the problem formalism in previous subsection), after $t$=$\tau$ the second type of problem was 20-10-5.

| Algorithm | Fixed | | | Probabilistic | | | SBM Network |
|---|---|---|---|---|---|---|---|
| | 20-20-5 | 20-10-5 | 20-5-5 | 20-20-5 | 20-10-5 | 20-5-5 | 25-30-35 |
| Modularity Max | 0.67±0.0 | 0.73±0.03 | 0.64±0.02 | 0.96±0.04 | **1.0±0.0** | **1.0±0.0** | **0.99±0.02** |
| Word2vec | 0.49±0.05 | 0.57±0.07 | 0.56±0.06 | 0.70±0.04 | 0.77±0.09 | 0.62±0.08 | 0.84±0.03 |
| MRIL | 0.25±0.09 | 0.38±0.12 | 0.36±0.11 | 0.43±0.14 | 0.39±0.07 | 0.24±0.04 | 0.46±0.10 |
| Original SyncMap | 0.93±0.12 | 0.75±0.08 | 0.63±0.11 | **1.0±0.0** | 0.81±0.04 | 0.64±0.08 | **1.0±0.0** |
| **Ours**: Symmetrical SyncMap | **1.0±0.0** | **1.0±0.0** | **0.93±0.08** | **1.0±0.0** | **1.0±0.0** | **1.0±0.0** | **1.0±0.0** |
| Algorithm | Mixed | | | Continual 15-15-5 to 20-10-5 | | | - |
| | 20-20-5 | 20-10-5 | 20-5-5 | Fixed | Prob. | Mixed | - |
| Modularity Max | 0.69±0.05 | 0.78±0.05 | 0.89±0.06 | 0.69±0.02 | 0.70±0.05 | 0.64±0.02 | - |
| Word2vec | 0.66±0.07 | 0.60±0.06 | 0.73±0.05 | 0.45±0.04 | 0.60±0.04 | 0.65±0.04 | - |
| MRIL | 0.20±0.05 | 0.20±0.05 | 0.16±0.03 | 0.38±0.13 | 0.59±0.02 | 0.55±0.04 | - |
| Original SyncMap | **0.84±0.08** | 0.83±0.0 | 0.64±0.07 | 0.72±0.07 | 0.82±0.04 | 0.83±0.0 | - |
| **Ours**: Symmetrical SyncMap | **0.87±0.09** | **0.90±0.06** | **0.95±0.04** | **1.0±0.0** | **1.0±0.0** | **0.95±0.06** | - |

Table 1: **NMI results.** A comparison is shown over Modularity Max, Word2vec, MRIL, original SyncMap and Symmetrical SyncMap in imbalanced and real-world CGCPs. The best and the non-statistically different results are in bold. Data are mean±s.t.d. Details of the statistical t-tests (p values) are presented in Appendix H.

**Results Overview.** The proposed algorithm Symmetrical SyncMap learns nearly the optimal solutions in all imbalanced CGCPs. It surpasses or ties other algorithms in all tests (Tables 1). Modularity Max performs the second best, in which it wins or ties the others in 2 out of 3 probabilistic CGCP tests. Word2vec achieves relatively higher NMI in probabilistic CGCPs than other problem structures, whereas MRIL performs the worst overall in all tests. The original SyncMap performs good in 20-20-5 CGCPs, yet performance decrease is witnessed as more chunks become smaller.

**Symmetrical SyncMap**, with its inherent adaptivity, performs significantly better than all other competitive algorithms, particularly in continual variations (i.e., dynamical CGCPs where environment can change). The proposed wider memory window and symmetrical activation allow capturing states in big chunks compactly, while at the same time the stochastic selection with suitable $Pr$ helps to separate small chunks (See the learned maps and $Pr$ analysis in Appendices D and F), thus keeping the balance between dealing with small and big chunks. In all probabilistic CGCP tests, it produces very distinct chunks and learns the best solution (i.e., NMI=1.0). The performance downgrades slightly during the more challenging mixed CGCP tests, due to an extra imbalanced frequency issue: the fixed chunk inserted between two probabilistic chunks has lower frequency to appear in the sequences. Having said that, the proposed algorithm still outperforms the others in all mixed CGCPs with a very big lead.

**Original SyncMap** performs relatively better in 20-20-5 type CGCPs, with the steady decrease in 20-10-5 and 20-5-5 ones. In fixed CGCPs, it makes distinct clusters for smaller chunks, yet fails to group nodes of the big chunk together. In probabilistic and mixed tests, nodes belong to smaller chunks are merged into one cluster in almost every individual trial (see Figures in Appendix D).

**Modularity Max** shares the highest NMI score with Symmetrical SyncMap in two probabilistic CGCP tests. However, this graph-based algorithm does not perform well in other imbalanced CGCPs. A possible reason is that fixed chunks are less likely to appear in usual problems faced by Modularity Max, thus leading to a problem bias. It is worth noting that comparing the results of Modularity Max to the other algorithms in dynamical CGCPs is not fair, since a TP matrix would record all the occurrence of state variables; thus, passing a continual changing TP matrix is not inherently suitable for Modularity Max, leading to worse results in continual structures than that in static graphs.

**Word2vec** creates maps in which nodes are more dispersed than that produced by SyncMap, thus making clustering difficult. It performs better in probabilistic chunk tests than fixed and mixed ones (see Figures in Appendix D). For the continual problems, Word2vec lacks the ability of adaptation, thus showing the overall lower NMI scores. **MRIL** fails to detect imbalanced chunks with large number of state variables, and therefore it performs the worst in all tests. Increasing the number of output neurons may improve the performance of fixed chunk tests.

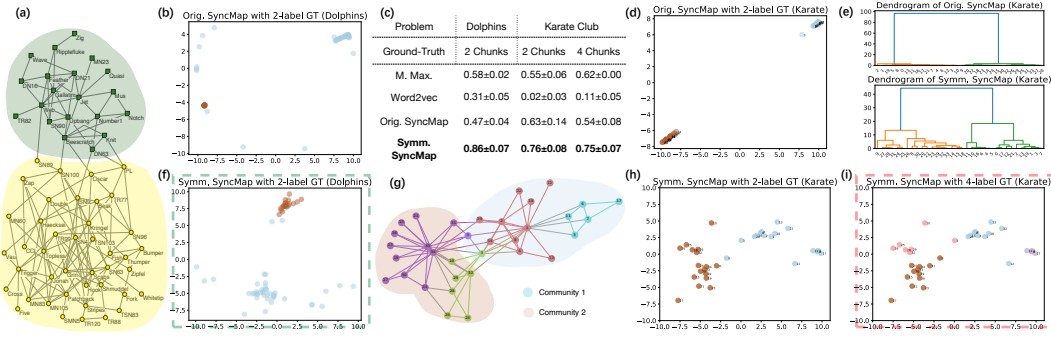

Figure 5: **Results of real-world scenarios with the original and Symmetrical SyncMap.** (a) Dolphins network. Colors denote labels. Figure modified from Arenas et al. (2008). (b) and (f) The learned representations of the two models (Dolphins network). (c) NMI results. (d) The learned representation of the original SyncMap (Karate network). (e) Dendrograms by hierarchical clustering (ward linkage) in Karate problems by two models. (g) Karate network. Colors of nodes denote local communities while colored shadow areas define the global communities. Figure modified from Perozzi et al. (2014). (h) and (i) The learned representations of Symmetrical SyncMap (Karate network). Colors indicate the true labels of the communities. See Appendix C for more analysis.

## 7 REAL-WORLD SCENARIOS

We study three real-world scenarios to verify the performance of Symmetrical SyncMap: (i) a network of stochastic block model (SBM); and (ii) two social network datasets with well-established community structures. For the SBM, we test a reference network introduced by Lee & Wilkinson (2019), where the network is considered as a graph which was then converted to a high-dimensional CGCP (i.e., 3 sightly imbalanced communities with a total of 90 nodes and 1192 edges). The ground-truth is defined in Appendix C. Parameter settings of all models remained the same as in previous imbalanced CGCP experiments. As shown in Table 1, both original and Symmetrical SyncMap yield the optimal solution, showing the capabilities to tackle with large-scale CGCP.

We then test two well-studied benchmark networks in community detection. Hierarchical clustering was applied to replace DBSCAN in the clustering phase, to produce dendrograms for the visualization of hierarchies (i.e., by specifying the number of communities/chunks). Detailed settings and analysis for baselines are in Appendix C. NMI results are shown in Figure 5(c) with statistics.

The first one is the Lusseau's network of bottlenose dolphins (Fortunato, 2010), an imbalanced structure with 2 ground-truth communities of sizes 20 and 42. Our algorithm yields a much higher NMI than other algorithms. It avoids forming dense communities produced by the original SyncMap (see Figure 5(b)(f)), allowing local relationships to be extracted, as verified in the following problem.

The second problem is the Zachary's karate club network which contains 34 nodes and 78 undirected and unweighted edges. We used two sets of ground-truth: (i) 2 chunks labeled by the original paper (Zachary, 1977), and (ii) 4 chunks found through modularity-based clustering (Perozzi et al., 2014).

Symmetrical SyncMap depicts the global graph structure while preserving the topology of local communities (Figure 5(e)(h)(i)). In contrast, the original SyncMap can only separate two global communities with a very dense representation (Figure 5(d)(e)). This unavoidable convergence is due to the stronger negative feedbacks over time, pulling away nodes from each community/chunk. Note that the representation learned by our method is comparable to node embeddings models with loss functions required and with more expensive training procedures such as DeepWalk (Perozzi et al., 2014) and Graph Convolutional Networks (Kipf & Welling, 2016). Unlike graph-based models, we achieve this by (i) mapping correlations from temporal input to a latent state space, (ii) keeping equilibrium by symmetrical activation (otherwise nodes would be locked in dense communities), and thus (iii) enabling hierarchies to be extracted from sequences. More importantly, the inherent adaptivity, as shown in previous experiments, suggests that our model a has potential usage in *inductive* applications, as it does not require any additional optimizations when dealing with new nodes/(sub)structures, while *transductive* methods such as DeepWalk cannot naturally generalize to unseen nodes or changed structures (Hamilton et al., 2017).

Having said that, we argue that these real-world scenarios usually (i) have no ground-truth and (ii) are strongly biased towards standard algorithms. To illustrate, the absence of ground-truth has to do with the fact that: it is not only difficult to define the social structures, but also hard to know the existence of real chunks in nature; thus, any answer would be a guess at most. Besides, the bias is due to the output, used as ground-truth, is found by using standard algorithms in the original papers, which makes good results in real-world data more like "algorithms that perform similar to standard algorithms", rather than "algorithms that work with real-world data".

## 8 CONCLUSIONS

We propose Symmetrical SyncMap, a brain inspired self-organizing algorithm built on top of the original work to solve continual general chunking problems (CGCP). Experiments of different CGCPs have illustrated how effective the concise modifications work on those challenging tasks. By applying symmetrical activation to the dynamical equations in which loss/optimization functions are not required, our algorithm not only learns imbalanced CGCP data structures with great long-term stability and adaptivity, but also shows the potentials to uncover complex hierarchical topologies encoded in temporal sequences. This reveals the self-organizing ability of the proposed algorithm in analyzing the community structures of a broad class of temporal inputs. Future goals, advised by the results presented in this paper, will be to investigate CGCP problems with large scale, hierarchies and noisy environments, and tasks specific to representation learning in various real-world scenarios.

## 9 REPRODUCIBILITY STATEMENT

We took every effort to make this work reproducible. Necessary codes are provided in supplementary materials, together with Python dependencies used to build the experiment environments. Please refer to the README file zipped in the supplementary material for detailed instructions.

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

## A    DEFINITION OF NORMALIZED MUTUAL INFORMATION (NMI)

Among the large number of clustering quality measurements, we used the Normalized Mutual Information (NMI) for measurement. Mathematically, NMI is defined as:

$$NMI(Y,\hat{Y}) = \frac{I(\hat{Y};Y)}{\frac{1}{2}(H(\hat{Y}) + H(Y))}, \quad NMI \in [0,1] \tag{5}$$

where $\hat{Y}$ and $Y$ are the output of algorithms and the truth labels, respectively. $I(\hat{Y};Y)$ is the mutual information and $H(*)$ is the entropy. NMI ranges between 0 and 1, and the higher the score, the better the clustering performance (better correlation between $\hat{Y}$ and $Y$).

## B    EXAMPLES OF IMBALANCED CGCPS

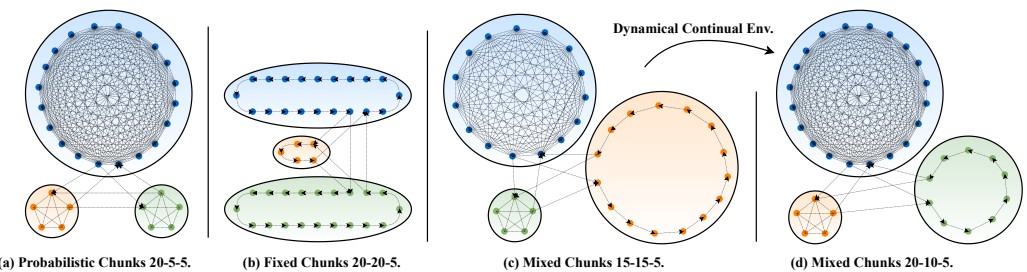

(a) Probabilistic Chunks 20-5-5.    (b) Fixed Chunks 20-20-5.    (c) Mixed Chunks 15-15-5.    (d) Mixed Chunks 20-10-5.

Figure 6: Examples of imbalanced chunking problems used in the experiments. CGCP generalizes several problems, (a) Probabilistic chunks: a graph structure that allows random walking; (b) Fixed chunks: temporal chunks defined originated from neuroscience; (c) and (d) Mixed chunks: integration of fixed and probabilistic chunks. In the continual setting, the causal structure can change over time. Dots inside each circle belong to a corresponding chunk. Lines connecting nodes without an arrow indicate that the transition is bidirectional; for directional transitions, arrows specify the direction.

## C    ANALYSIS OF REAL-WORLD SCENARIOS

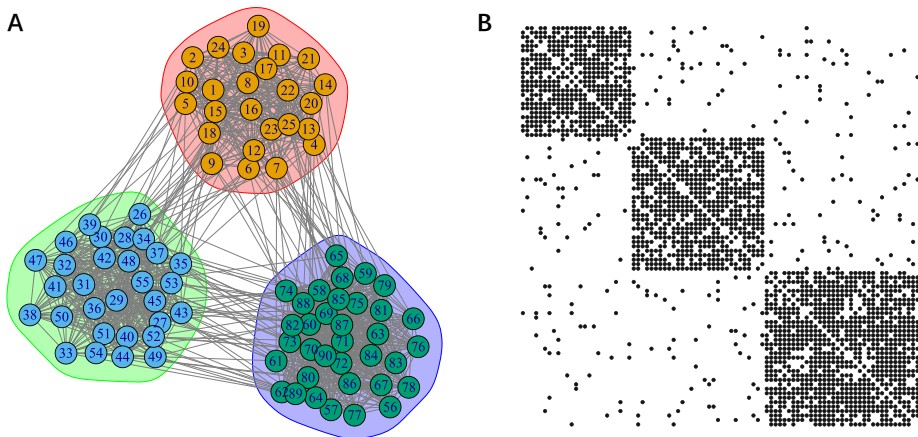

Figure 7: Stochastic block model (SBM) network used in experiments. (a) Graph introduced in Lee & Wilkinson 2019, which consists of 90 nodes and 1192 edges. The nodes are divided into 3 groups, with groups 1, 2 and 3 containing 25, 30 and 35 nodes, respectively. The nodes within the same group are more closely connected to each other, than with nodes in another group. The connectivity of the nodes is considered uniform transition. (b) Corresponding adjacency matrix for graph in (a).

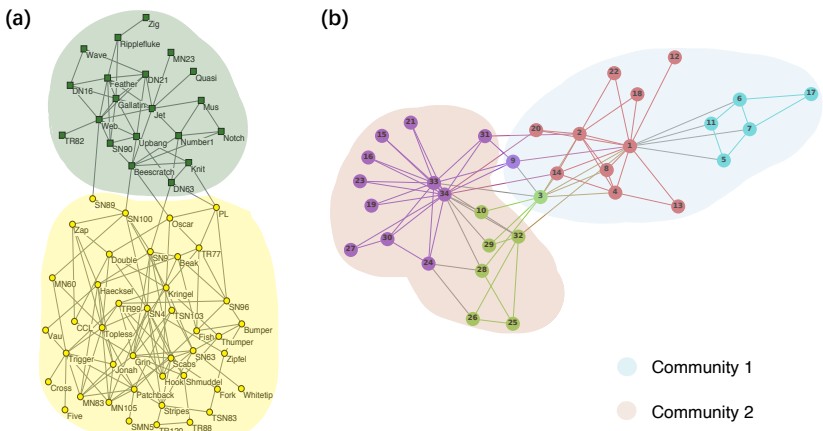

Figure 8: Two community detection benchmarks used in experiments. (a) The Lusseau's network of bottlenose dolphins (Fortunato, 2010), modified from (Arenas et al., 2008). This network is considered as an imbalanced structure with 2 ground-truth communities of sizes 20 and 42. (b) The Zachary's karate club network, modified from (Perozzi et al., 2014). This contains 34 nodes and 78 undirected and unweighted edges. Colors within nodes denote local communities (i.e., the ground-truth found by the modularity-based algorithm (Perozzi et al., 2014)), while colored shadow areas define the global communities (i.e., the ground-truth collected from the original paper (Zachary, 1977)).

## C.1 EXPERIMENT SETTINGS OF MODELS USED IN REAL-WORLD SCENARIOS

Here, we specify the detailed settings in the experiments of real-world problems.

**Stochastic block model (SBM) network.** The structure of SBM network is shown in Figure 7. As mentioned in the main text, all settings remained the same as in previous imbalanced CGCP experiments (see Section 6.2).

**Community detection benchmarks.** In the two problems of the community detection benchmarks, MRIL was not used, since we focus on the investigation of how the given algorithms learn the topological and hierarchical structures underlying in input sequences, as well as how well the structures are produced using such algorithms. MRIL cannnot encode input sequences into a map space, therefore it was not considered as a baseline.

Regarding the Modularity Max, we again used TP matrices that produced by the graphs generated from the given input sequences, as our focus is on the ability of extracting information in input sequences. When finding the communities, we specify the "number of communities" to the algorithm. This can be found in the official documentation of NetworkX (a Python library for analyzing graphs), where we set "a minimum number of communities below which the merging process stops. The process stops at this number of communities even if modularity is not maximized." However, it should be noted that the process will stop before the cutoff if it finds a maximum of modularity. Based on the given ground-truth, in the Lusseau's network of bottlenose dolphins, we set the "number of communities" at 2. In the Zachary's karate club network, we set the "number of communities" at 2 and 4 for computing NMI with two sets of ground-truth, respectively.

Regarding the Word2vec, we set the latent dimension equal to 2, and kept all other parameter settings unchanged. This is to produce a 2-D representation of the given community structures. Also, in the previous experiments, DBSCAN was used to obtain chunks, where in the real-world problems we replaced it to hierarchical clustering as we are more interested in the topology as well as the hierarchical structures.

The original SyncMap and Symmetrical SyncMap shared the same changes with Word2vec, that is, we only reduced the SyncMap space dimension $k$ from 3 to 2. Also, hierarchical clustering is used.

Regarding the hierarchical clustering, we used "ward" as a linkage method. And we specified the number of clusters when performing the algorithm. In details, for the Lusseau's network of bottlenose dolphins, we set the "number of clusters" at 2. In the Zachary's karate club network, we set the "number of clusters" at 2 and 4 for computing NMI with two sets of ground-truth, respectively.

## C.2 Results Analysis

**SBM network.** Modularity Max performs nearly optimal in this scenario. Also, recall that it yields relatively low NMI in fixed CGCP. The differences of the performance observed here might be because the communities with many deterministic connections are less likely to appear in usual problems faced by Modularity Max, thus leading to a problem bias; that is, fixed chunk structures are not strictly meet the condition that the possibility of transition to an internal state within a chunk is higher than that of transition to an external state belonging to other chunks. This leads to a problem bias for Modularity Max.

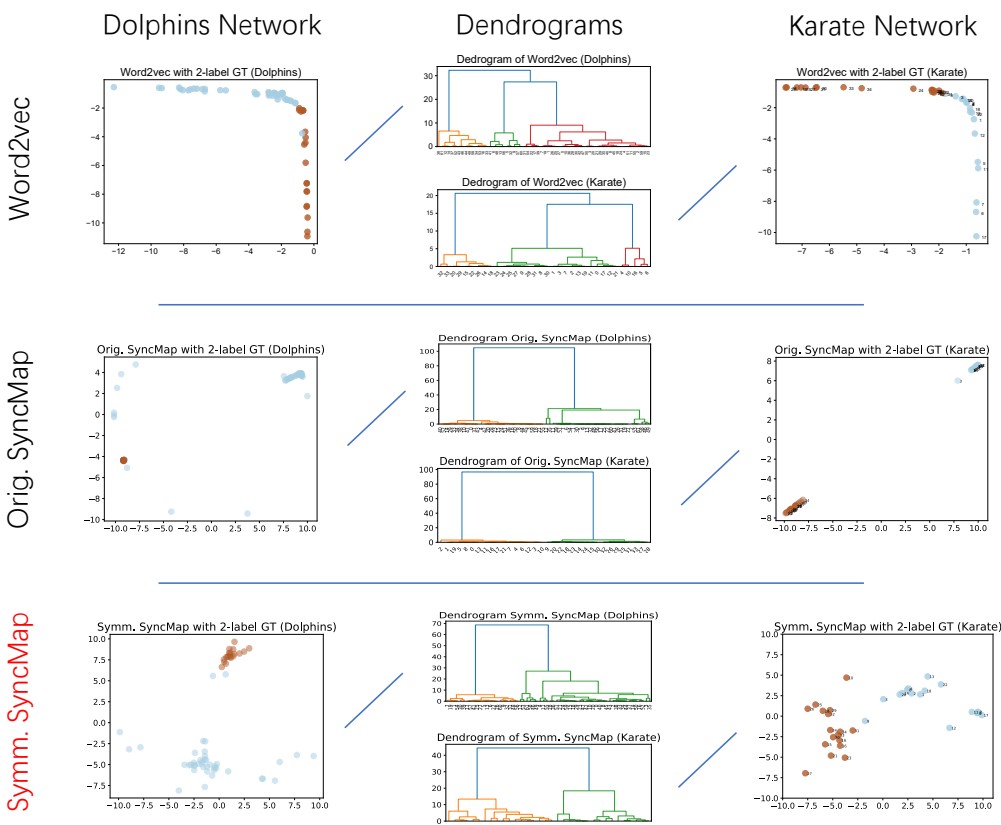

Figure 9: Analysis of real-world problems used in the experiments. Here we compare three algorithms, namely (i) Word2vec, (ii) original SyncMap, and (iii) Symmetrical SyncMap (the proposed one). Note that Modularity Max and MRIL do not encode input sequences into a map space. Thus, they are not figuratively comparable.

**Community detection benchmarks.** Word2vec, as shown in Figure 9, always forms a latent representation with two long tails. This shape does harm when finding hierarchical structures.

We have analyzed the two models of SyncMap in the main text. It is worth noting here that from the dendrograms of the original SyncMap, we can conclude that the original work is not possible to extract local relationships (i.e., hierarchies), as the nodes are all densely distributed in compact communities.

Regarding the Modularity Max, it does not encode temporal sequences into a map space, and thus it is not figuratively visualized. Meanwhile, although the datasets here are designed for modularity-based

models, **Modularity Max** failed to extract information and maximize the modularity from sequential data, thus showing lower NMI.

## D  MAPS LEARNED BY DIFFERENT ALGORITHMS

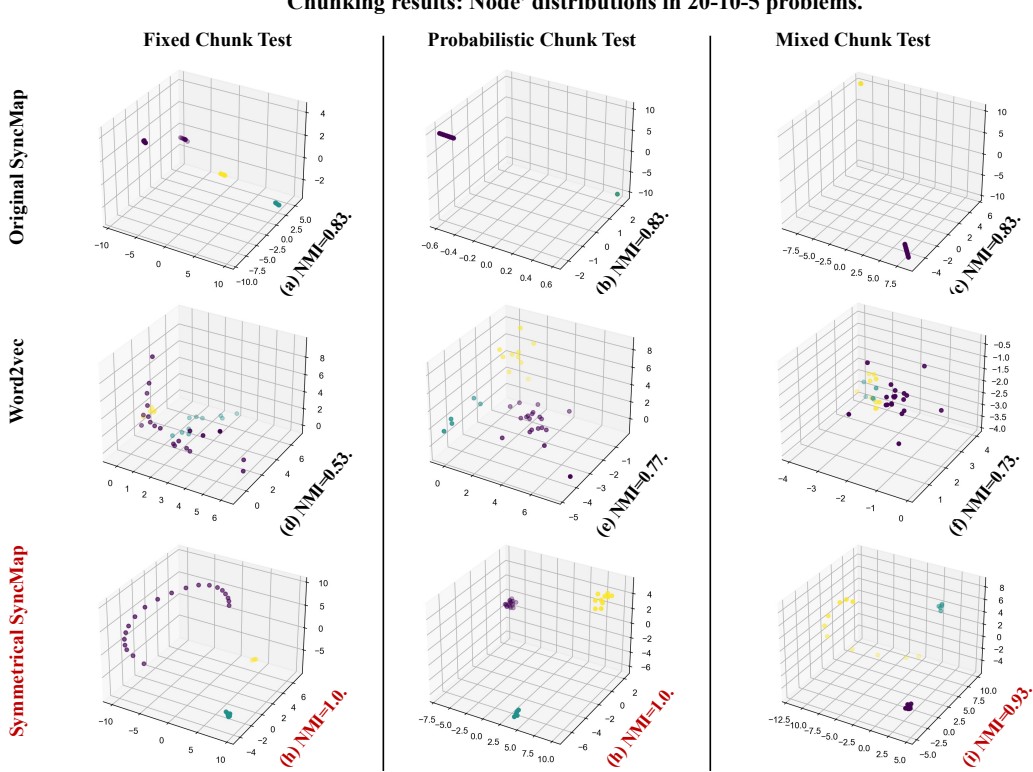

Figure 10: Chunking results of imbalanced 20-10-5 fixed (left), probabilistic (middle) and mixed (right) problems using the original SyncMap (top), Word2vec (middle) and the proposed Symmetrical SyncMap (bottom). Colors of the nodes indicate the true labels of chunks. Specifically, purple for big chunk (20), yellow for moderate chunk (10), green for small chunk (5). Note that Modularity Max and MRIL do not encode input sequences into a map space. Thus, they are not figuratively comparable.

## E  STABILITY ANALYSIS

The proposed Symmetrical SyncMap shows good stability over the long run in all imbalanced chunking problems. Figures 11 and 12 show the NMI average with error bar (s.t.d.) at every 10,000 time step during training in all imbalanced problems.

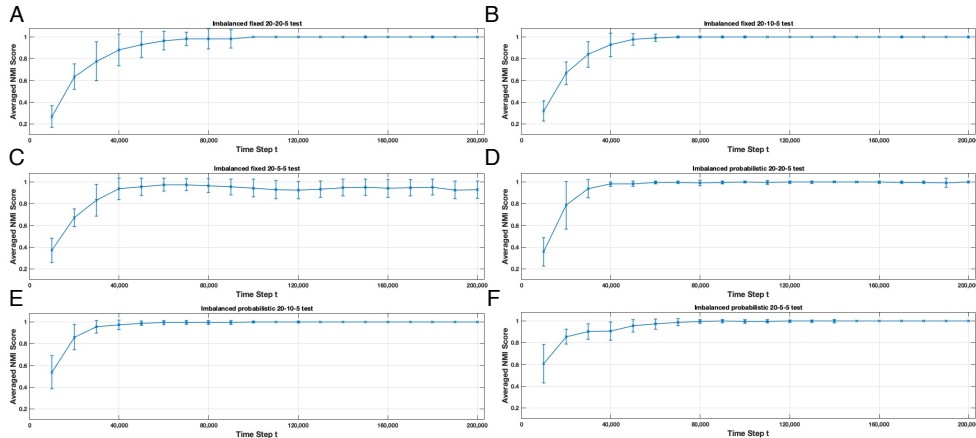

Figure 11: NMI over time of imbalanced tests. (a) fixed 20-20-5, (b) fixed 20-10-5, (c) fixed 20-5-5, (d) probabilistic 20-20-5, (e) probabilistic 20-10-5, (f) probabilistic 20-5-5.

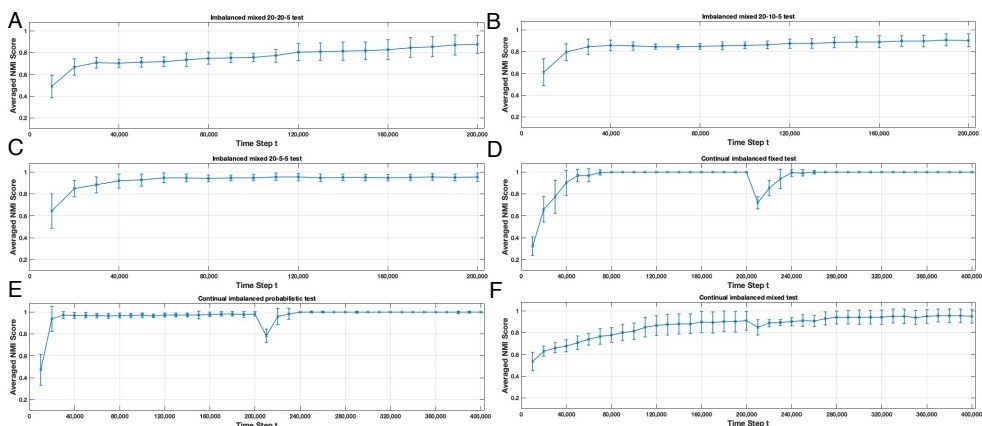

Figure 12: NMI over time of imbalanced tests. (a) mixed 20-20-5, (b) mixed 20-10-5, (c) mixed 20-5-5, (d) continual fixed, (e) continual probabilistic, (f) continual mixed.

## F  PARAMETER SENSITIVITY ANAYLSIS

Table 4 shows the performance of Symmetrical SyncMap on the same imbalanced experiments (fixed/probabilistic/mixed 20-10-5 test) but with different parameters settings. Results suggest that Symmetrical SyncMap is robust to changes in parameters, with mostly smooth changes.

Note that the parameter $Pr$ is designed for solving small chunks. To illustrate, if $Pr = 100\%$, then at every time step we apply state memory generalization to have a longer memory window. In this case, if the current chunk is too small (e.g., only having 3 states), then it would probably not be detected. Therefore, choosing $Pr = 30\%$ provides a trade-off between finding big and small chunks. In fact, in the future version of our model, this parameter will no longer be used, as tiny chunk/communities are rarely appear alone, while they usually appear inside a big chunk (i.e., hierarchies). Having said that, the results of using $Pr = 100\%$ and $Pr = 30\%$ are both adequate.

Table 2: NMI of several Symmetrical SyncMap variations in Fixed, Probabilistic and Mixed structures settings (20-10-5 tests).

| Setting | | | Fixed | Probabilistic | Mixed |
|---|---|---|---|---|---|
| k = 3 | m = 3 | Pr = 30% | 1.0±0.0 | 1.0±0.0 | 0.92±0.06 |
| k = 3 | m = 2 | Pr = 100% | 0.95±0.08 | 0.98±0.02 | 0.86±0.03 |
| k = 3 | m = 3 | Pr = 100% | 1.0±0.0 | 1.0±0.0 | 0.95±0.05 |
| k = 3 | m = 4 | Pr = 30% | 0.96±0.08 | 1.0±0.0 | 0.95±0.05 |
| k = 2 | m = 2 | Pr = 100% | 0.90±0.10 | 1.0±0.0 | 0.87±0.07 |
| k = 2 | m = 3 | Pr = 30% | 0.98±0.07 | 1.0±0.0 | 0.90±0.08 |
| k = 2 | m = 4 | Pr = 30% | 0.95±0.08 | 1.0±0.0 | 0.94±0.07 |

## G  ALGORITHMIC DESCRIPTION OF SYMMETRICAL SYNCMAP

---

**Algorithm 1** Symmetrical SyncMap

---

**Input**: input sequence $\boldsymbol{X} = \{\boldsymbol{x_i}|i = 1, ..., \tau\}$
**Parameters**: sequence length $\tau$, map dimension $k$, state memory $m$, probability parameter $Pr$, input dimension $n$, learning rate $\alpha$
**Output**:     A     number     of     clusters     indicating     communities     and chunks.

1: Initialize SyncMap by generating weight nodes $w_{i,0}$ , where $i = 1...n$
2: Set $W_t = \{w_{i,t}|i = 1...n, t = 0...\tau\}$
3: **for** $t = 0$ to $\tau$ **do**
4:     Initialize $PS_t$ and $NS_t$ as empty set
5:     Randomly generate a constant variable $Pr_t \in [0, 1]$
6:     Divide $m$ nodes into temporary set $PS_{temp}$
7:     $m_{neg} \leftarrow m$ (Symmetrical activation)
8:     **if** $m > 2$ & $Pr_t < Pr$ **then**
9:         Stochastically select 2 nodes in $PS_{temp}$ to activate
10:         Include these 2 nodes into set $PS_t$
11:         $m_{neg} \leftarrow 2$ (Symmetrical activation)
12:     **else if** $m > 2$ & $Pr_t \geq Pr$ **then**
13:         $PS_t \leftarrow PS_{temp}$
14:     **end if**
15:     Set temporary $NS_{temp} \leftarrow W_t - PS_t$
16:     Stochastically select $m_{neg}$ nodes in $NS_{temp}$ to activate
17:     Include these $m_{neg}$ nodes into set $NS_t$
18:     $PS_t$ and $NS_t$ determined
19:     Calculate $cp_t$ and $cn_t$ by Eq. 2
20:     Update the nodes' position by Eqs. 3 and 4
21:     Normalize nodes in hypersphere radius $= 10$
22: **end for**
23: Apply clustering algorithm such as DBSCAN or Hierarchical clustering

---

## H  STATISTICAL TESTS

We used a t-test with p-value of 0.05 to verify if the best result is statistically significantly different from other results. $h$ is the hypothesis test result ($h$=0 indicates a failure to reject the null hypothesis at the 5% significance level, and $h$=1 otherwise). $p$ is the two-tailed $p$ value, and $ci$ is the confidence interval for the difference in population means of two samples.

Table 3: Statistical Results.

| Problems | Description | Num. of Samples and Mean+s.t.d. | Test | Statistic |
|---|---|---|---|---|
| Long-term Analysis (Figure 5) | Orig. vs Symm.SyncMap | 30 each model at final time step | Two-sample t-test | h=1, **p=6.5228e-27**, ci=[0.3166;0.3899] |
| Prob. 20-20-5 (Table 1) | M.Max vs Symm.SyncMap | 30 each model (0.96±0.04 and 1.0±0.0) | Two-sample t-test | h=1, **p=4.3546e-06**, ci=[0.0220;0.0508] |
| Prob. 20-20-5 (Table 1) | Orig. vs Symm.SyncMap | 30 each model (1.0±0.0 and 1.0±0.0) | Two-sample t-test | h=NaN, **p=NaN**, ci=[0.0;0.0] |
| Prob. 20-10-5 (Table 1) | M.Max vs Symm.SyncMap | 30 each model (1.0±0.0 and 1.0±0.0) | Two-sample t-test | h=0, **p=0.3215**, ci=[-0.0033;0.0098] |
| Prob. 20-5-5 (Table 1) | M.Max vs Symm.SyncMap | 30 each model (1.0±0.0 and 1.0±0.0) | Two-sample t-test | h=0, **p=0.3216**, ci=[-0.0042;0.0125] |
| Mixed 20-20-5 (Table 1) | Orig. vs Symm.SyncMap | 30 each model (0.84±0.08 and 0.87±0.09) | Two-sample t-test | h=0, **p=0.1568**, ci=[-0.0121;0.0731] |
| SBM Network (Table 1) | M.Max vs Symm.SyncMap | 30 each model (0.99±0.02 and 1.0±0.0) | Two-sample t-test | h=0, **p=0.09**, ci=[-0.0011;0.0129] |
| SBM Network (Table 1) | Orig. vs Symm.SyncMap | 30 each model (1.0±0.0 and 1.0±0.0) | Two-sample t-test | h=NaN, **p=NaN**, ci=[0.0;0.0] |

## I  COMPUTATIONAL TIME ANAYLSIS

We analyze the computational time over Symmetrical SyncMap, SyncMap and Word2vec. In addition to Karate network, we introduced two larger scale imbalanced CGCP problems (i.e., problem with 300-D input and 1200-D input). More specifically, the structure of 300-D CGCP problem includes: Four chunks with 50 states + Two chunks with 20 states + Four chunks with 10 states + Four chunks with 5 states. We denote this problem as 300-D (50x4 + 20x2 + 10x4 + 5x4). Using the similar denotation, the 1200-D CGCP problem is 1200-D (300x2 + 150x2 + 50x4 + 20x2 + 10x4 + 5x4).

We ran all three problems 10-time per problem per algorithm with 200,000 as sequence length. All tests were run on a MacBook Pro 2.4GHz Quad-Core Intel Core i5 16GB laptop as they demand little computational effort, and computation time [second] were obtained in mean±s.t.d. Results show that the proposed Symmetrical SyncMap is scalable. Although it is slower than the original one and Word2vec (mainly due to the stochastic selection process), the computation time does not become worse as the scale of the input increases. Note that both SyncMap and the proposed algorithm should improve considerably if parallelization, GPU programming and other techniques are employed. For example, all nodes in SyncMap can be updated at the same time.

Table 4: Computation time[s] comparison over Symmetrical SyncMap, SyncMap and Word2vec.

| Problem Type | Symm. SyncMap | Orig. SyncMap | Word2vec |
|---|---|---|---|
| Karate (34-D) | 90.227±1.297 | 30.179±0.571 | 33.077±0.429 |
| Imbalanced 300-D | 98.055±3.689 | 34.462±0.671 | 43.035±0.494 |
| Imbalanced 1200-D | 118.602±1.629 | 44.827±0.979 | 72.956±1.923 |

## J  ABLATION STUDY

The proposed method is composed of two parts compared to the original SyncMap: Symmetrical activation and genalized memory window. We perform an ablation study to evaluate the effect of each of these two modifications. The ablation study investigated CGCP problem of mixed-20-10-5. The key feature of this problem is that there is a fixed chunk with 10 state in between two probabilistic chunks. The 10-time averaged NMI results are shown in Table 5.

From the NMI scores, it is hard to image what happen inside the SyncMap space. However, a better visualization can be observed in the learned map comparison in Figure 13. To illustrate briefly, if

Table 5: NMI score of ablation study.

| Model Type | NMI Score |
|---|---|
| Original SyncMap (m=2) | 0.84±0.07 |
| SyncMap with Generalized Memory Window (m=3) | 0.83±0.0 |
| SyncMap with Symmetrical Activation (m=2) | 0.84±0.03 |
| Symmetrical SyncMap (m=3) | 0.90±0.06 |

only applying symmetrical activation, the states (i.e., weight nodes in the map) of the given fixed chunk will be very sparely distributed (Figure 13(c), i.e., the system cannot detect the fixed chunk because it needs more memory to remember fixed chunks). On the other hand, if only applying generalized longer memory window, the strength of the negative feedback will still dominate the self-organizing process, thus resulting in the similar output with Original SyncMap (Figure 13(a)(b); that is, all weight nodes are squeezed into compact clusters. Therefore, by applying symmetrical activation and generalized memory window together, the system is stable and able to detect fixed chunk in more general cases (Figure 13(d).

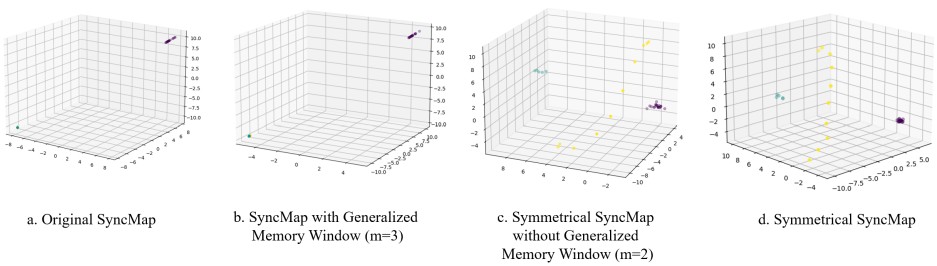

a. Original SyncMap    b. SyncMap with Generalized Memory Window (m=3)    c. Symmetrical SyncMap without Generalized Memory Window (m=2)    d. Symmetrical SyncMap

Figure 13: Learned map of (a) Original SyncMap, (b) SyncMap with Generalized Memory Window (m=3), (c) Symmetrical SyncMap without Generalized Memory Window (m=2), and (d) Symmetrical SyncMap. Note the nodes belong to fixed chunk is denoted by yellow color, which is slightly hard to visualize.

## K   RELATED WORKS

Chunking is an extremely multidisciplinary problem. In the main text, we attempt to cover a number of related topics from neuroscience to machine learning and try to connect them to this work. Due to the page limit, we review additional related works here for completeness.

**Latent Variable Estimation.** Some literature focusing on latent variable estimation solve problems which are related to chunking (Fox et al., 2011; Qian & Aslin, 2014; Pfau et al., 2010). However, they have different objectives, since chunking is a self-organizing process over the variables of the problem with respect to their temporal correlation. Even if chunks of variables can be abstracted as a set of variables, there is still an inherent difference between chunks and latent variables.

**Unsupervised Learning for Sequences.** Feature extraction is commonly used to predict future input in unsupervised learning for sequences (Clark et al., 2019; Lei et al., 2017; Mikolov et al., 2013; Wu et al., 2018). A peculiar learning algorithm called contrastive predictive encoding (CPE) has been proposed, which represents sequences by using a probabilistic contrastive loss in the latent space to encode maximally useful information (Hjelm et al., 2018). CPE solves problems which are complementary rather than competing with the problem presented here.

