# OpenReview forum: "Symmetrical SyncMap for Imbalanced General Chunking Problems"
_ICLR.cc/2023/Conference — Submitted to ICLR 2023_

### Official Review · Reviewer_SjDm · 2022-10-22

**Confidence:** 3
**Clarity, Quality, Novelty And Reproducibility:** Have met the standards.
**Correctness:** 3
**Technical Novelty And Significance:** 3
**Empirical Novelty And Significance:** 3
**Recommendation:** 5

**Strength And Weaknesses:**

Pros:

- The paper is clearly written and well organized, the formulation of the problem is good, figures and descriptions help in understanding CGCP and SyncMap.
- The idea of introducing even node updating is intuitive and interesting (also effective in results).
- Competitive performance to SOTA methods, especially to vanilla SyncMap, and the results are even 1 in most cases.
- Real-world scenarios evaluation helps understanding of the method, settings and formulations are clear.

Cons:

- CGCP is a newly formulated problem, while few comparison methods are evaluated on this issue. Though this paper has listed some conventional and SOTA methods, there is still some space for comparison.
- The author mentioned, “As opposed to the original SyncMap that suffers from the uneven updates …” in the Intro, and “the dynamic’s update is more influenced by negative feedback loop,” in Limitations of SyncMap. Though the motivation is intuitive that unbalanced negative samples may bring extra catastrophic information to the training phase, there do exist situations when there are more positive samples, and in that way, maybe even sampling strategy is no good as uneven does.
- This novel method is composed of two parts compared to the vanilla SyncMap: symmetrical activation and memory window. I’m concerned that if the authors have evaluated the effect of each of these two components, i.e., does one of the components bring a big boost, or is it the combination of these two that helps much? The integration of both demands further explanation.

Minor:

- Extra “)” in Training Dynamic, page 4.


**Summary Of The Paper:**

This paper first identifies problems of SyncMap in Continual General Chunking Problem (CGCP) with long-term instability and poor performances given the asymmetric number of updates of nodes. To solve these issues, a variant of SyncMap, Symmetrical SyncMap is proposed using a symmetric selection of nodes, and a generalized memory window to avoid suffering from the uneven updates from positive/negative feedback loops, and helps the system have more updates from the positive feedback loop, respectively. The authors also show comparisons in 12 imbalanced CGCPs with various difficulties, indicating the proposed method surpasses or ties with other unsupervised SOTA baselines. The authors also verify the performance in real-world scenarios.



**Summary Of The Review:**

Overall, I like this method, also the problem CGCP generated from neuroscience to computer science. I can see the potential of this problem and this method. The motivation, settings, and evaluations are clear, and all else aside, I felt a lot of insights from this paper. However, I'm just yet convinced by the current amount and set of experiments, also the ablations.

---

> ### Author Response · Authors · 2022-11-14
> **Thank you for your review**
>
> Thank you for a thorough review and questions that will help us improve the manuscript. We will take your comment into account in a final version of the paper, including all the minor issues. Please see below for the discussion.
>
> (1) Ablation study:
>
> We are aware of the lack of ablation study of how symmetrical activation and generalized memory window improve the performance individually. Here we conducted an ablation study by investigating CGCP problem of mixed-20-10-5. The key feature of this problem is that there is a fixed chunk with 10 state in between two probabilistic chunks. The 10-time averaged NMI results are shown below.
>
> ==Mixed 20-10-5:
>
> =====Applying symmetrical activation (m=2) only: 0.84±0.03
>
> =====Genializing memory window to m=3 only: 0.83±0.0
>
> =====Applying both (i.e., the proposed one): 0.90±0.06
>
> From the NMI scores, it is hard to image what happen inside the SyncMap space. We will include a learned map (similar to Appendix D) for better visualization in the final version of the paper.
> To illustrate briefly, if only applying symmetrical activation, the states (i.e., weight nodes in the map) of the given fixed chunk will be very sparely distributed (i.e., the system cannot detect the fixed chunk because it needs more memory to remember fixed chunks).
> On the other hand, if only applying generalized longer memory window, the strength of the negative feedback will still dominate the self-organizing process, thus resulting in the similar effect shown in Figure 10 (top-right); that is, all weight nodes are squeezed into compact clusters.
> Therefore, by applying symmetrical activation and generalized memory window together, the system is stable and able to detect fixed chunk in more general cases.
>
> (2) Cases when there are more positive samples than negative:
>
> In our setting, all states in the input sequence are divided into negative and positive samples based on their recent activity. Thus, it is unlikely that “over 50% of state variables are recently activated”. That is to say, the number of positive samples at each time step will stay in a very low portion of the total. Taking reviewer’s comment into account, we will take a deep investigation of this issue to make the algorithm robust given some extreme cases.
>
> (3) There are still some space for comparison:
>
> With regard to previous work, we were not aware of methods in different disciplines that can be used to tackle CGCP problems. For example, as pointed out by Reviewer 7tpF, Stochastic block model based algorithm such as BLOck-wise Sbm learning (BLOS) published in 29th AAAI shows great potential to solve CGCP (the NMI scores for Karate and Dolphin networks of BLOS are 0.839 and 0.66, compared with that of ours of 0.76 and 0.86). Additionally, a simple (Auto-Regressive) Hidden Markov Model would likely uncover the "state variables" defined in the paper, but not the
> "chunks" of state variables that we seek to uncover.
> However, hierarchical versions of such models would in fact find some version of "chunks" (i.e. sequences of states). A recent (potentially) relevant paper in this regard is "Variational Predictive Routing with Nested Subjective Timescales" by Zakharov et al. (https://arxiv.org/pdf/2110.11236.pdf); (also see the section "Hierarchical temporal structure" in their Related Work for more examples).
>
> We will incorporate these citations in our paper and rephrase some parts to give due credit to these researchers. Simply put, these works strengthen our belief that our work is indeed relevant, as we have seen many different incarnations of the CGCP problem. These research lines imply our model could indeed be used to model more challenging sequential data.

---

### Official Review · Reviewer_hkor · 2022-10-23

**Confidence:** 3
**Correctness:** 4
**Technical Novelty And Significance:** 3
**Empirical Novelty And Significance:** 3
**Recommendation:** 5

**Clarity, Quality, Novelty And Reproducibility:**

The paper is well written. It is easy for people not in the domain to easily understand the question and the idea.
However, many figures are too small and hard to read into details.
All experiments are provided with code and reproduction detail.

**Strength And Weaknesses:**

Strengths:
1. The proposed method is well inspired by brain activity. It identifies the limitation of the previous syncmap algorithm and improves the algorithm with better sampling. It also proposes sampling tricks such as memory windows.
2. The empirical advantage of the proposed method is strong. It outperforms previous methods.
3. The balancing trick and memory window idea can be useful for other domains, e.g., contrastive learning. Hence, this paper may have an impact to out-of-domain topics.

Weaknesses:
1. All the experiments are conducted in a relatively small scale, including both model size and data size. It is questionable how fast this algorithm can converge. As a non-deep-learning algorithm, it is recommended to discuss convergence speed and overall computation/memory complexity.
2. In addition, the discussion over 'Symmetrical SyncMap depicts the global graph structure while preserving the topology of local communities' seems not well supported. This further limits the scope that the proposed algorithm can apply to.

**Summary Of The Paper:**

This paper proposes an improvement over the sync-map algorithm for imbalanced continual general chunking problems (CGCPs). The problem is effectively clustering on sequential data. The original sync-map samples positive pairs and negative nodes (any number), pull together positive pairs, and push apart negative nodes. The improvement is to only sample the same number of negative nodes. In addition, in order to find these negative pairs, the authors propose stochastic selection and memory window.
Experimental results demonstrate the superior performance of the proposed methods over previous methods like Modularity Max, MRIL, and Word2vec.

**Summary Of The Review:**

An effective improvement over syncmap algorithm for imbalance general chunking problems. The paper has limited scope as it's only proposed to solve these problems and not seem to be scalable.

---

> ### Author Response · Authors · 2022-11-14
> **Thank you for your review**
>
> Thank you for a thoughtful review and questions. We will take your feedback onboard as we prepare a final version of the paper, including re-designed larger figures for better readability.
>
> We were not aware of conducting time analysis previously, we will take reviewer’s comment into account by introducing a comprehensive time analysis in the Appendix in the final version of the paper.
>
> Here, as for the discussion, we performed a quick analysis of computational time comparison over Symmetrical SyncMap, SyncMap and Word2vec. We introduced two larger scale problems (i.e., problem with 300-D input and 1200-D input), as shown below. We ran all three problems 10-time per problem per algorithm with 200,000 as sequence length. All tests were run on a MacBook Pro 2.4GHz Quad-Core Intel Core i5 16GB laptop as they demand little computational effort, and computation time [second] were obtained in mean±s.t.d.
>
> ==Karate network (dimension: 34-D, small)
>
> =====Symmetrical SyncMap: 90.227±1.297
>
> =====Original SyncMap: 30.179±0.571
>
> =====Word2vec: 33.077±0.429
>
> ==Imbalanced CGCP with 300-D input (50x4 + 20x2 + 10x4 + 5x4): Four chunks with 50 states + Two chunks with 20 states + Four chunks with 10 states + Four chunks with 5 states
>
> =====Symmetrical SyncMap: 98.055±3.689
>
> =====Original SyncMap: 34.462±0.671
>
> =====Word2vec: 43.035±0.494
>
> ==Imbalanced CGCP with 1200-D input (300x2 + 150x2 + 50x4 + 20x2 + 10x4 + 5x4):
>
> =====Symmetrical SyncMap: 118.602±1.629
>
> =====Original SyncMap: 44.827±0.979
>
> =====Word2vec: 72.956±1.923
>
> Results show that the proposed Symmetrical SyncMap is scalable. Although it is slower than the original one and Word2vec (mainly due to the stochastic selection process), the computational time does not become worse as the scale of the input increases. Note that both SyncMap and the proposed algorithm should improve considerably if parallelization, GPU programming and other techniques are employed. For example, all nodes in SyncMap can be updated at the same time.
>
> Regarding the scale of the problems, in this work, we mainly focus on identifying the existing issues in the original SyncMap and propose simple but efficient improvements, making the algorithm stable and prepared to deal with wide-range of real-world problems. Therefore, we start with small-scale CGCP problems, adding one at a time. In fact, in our other ongoing works, we are able to use Symmetrical SyncMap to tackle high-dimensional CGCP problems up to 1200-D with a slight modification of the update equation (i.e., NMI over 0.95), also we can apply Symmetrical SyncMap to problem up to 4000-D for image processing (64-by-64 resolution) with adequate results.

---

### Official Review · Reviewer_7tpF · 2022-10-26

**Confidence:** 2
**Clarity, Quality, Novelty And Reproducibility:** please see above
**Correctness:** 4
**Technical Novelty And Significance:** 2
**Empirical Novelty And Significance:** 2
**Recommendation:** 5

**Strength And Weaknesses:**

Strength

The paper studies an interesting problem and its proposed method seems to be working well in multiple synthetic and real-world settings.

Weakness

The technical contribution seems incremental. It turns a hard-code value of SyncMao into a hyperparameter and forces the positive set and negative set to be equally large. Both seem very minor tweak on the original SyncMap method.

Moreover, they use a stochastic way to match the size of positive and negative sets, which introduces extra noise. I am concerned with it since the goal of this work is to make the original SyncMap method more stable. Although the experiments have shown that their method is better than original SyncMap, I would still like to see a more in-depth stability analysis that discusses why introducing this new noise is not an issue.

The technical definitions for fixed chunks and prob chunks are a little strange to me. It seems that all chunks are probabilistic but some of them have no branching structures inside them so the internal transition probability is always 1.

I am also concerned with their evaluation.
From my understanding of this problem setting, I think stochastic block model seems a natural baseline method. Actually, the authors used SBMs to synthesize some of their experiments. Then why not use SBM to learn this community structure?

Another reasonable baseline is to learn a markov model whose transition probability matrix is strongly regularized: it is sparse; it forms local high-prob structure. Why don't the authors consider baseline methods of this style?

Writing needs improvements.

The technical part of the paper is not very self-contained. E.g., it is not clear how the positive set and negative set (or even eqn-2,3,4 in general) are related to the later stage of clustering. The clustering phase is introduced by simply citing DBSCAN without specifying how it works in this particular technical setting.

There are typos and errors. E.g., "positive (7) nodes and negative (2) nodes" is not consistent with the figure.



**Summary Of The Paper:**

This paper aims to address the problem of learning complex structures from sequences.
They improved an existing method called SyncMap by balancing the number of updates from positive and negative feedback loops.
They demonstrated the effectiveness of the proposed method on multiple synthetic and real-world datasets.

**Summary Of The Review:**

The paper address an interesting problem and proposed a method that works well in their experiments.
But I have concerns about their technical designs, experiments, and writing, so I tend to reject the paper.

---

> ### Author Response · Authors · 2022-11-14
> **Thank you for your review**
>
> Thank you for an insightful review and feedback, we will take this with us as we revise the manuscript. Please see below for answers to your questions.
>
> (1) Stochastic selection might introduce extra noise:
>
> The unstable effect in the original SyncMap, as identified in the paper, is due to the unbalanced updates in positive and negative feedback loops. By using symmetrical activation, we introduce a type of regularization of the forces (otherwise problems with more state variables would have stronger negative force). This regularization is to select state variables at every step that are already sampled by the same process in the original SyncMap (e.g., select 3 out of 20 negative state variables). In this scenario, no extra noise is introduced, and the stochastic selection is kind of a procedure to slow down the negative convergence process, which helps the system to reach equilibrium. More stability analysis results are shown in the Appendix E, in which Symmetrical SyncMap are stable for all problems.
>
> (2) Technical definition of fixed chunk seems to be a special type of probabilistic chunk:
>
> Reviewer is right, that the fixed chunk has the internal probability at 100% when transition between states in the same chunk. We highlight fixed chunk as a problem different from the probabilistic one, because fixed chunk is originated from neuroscience while probabilistic chunk is more related to community structures. We will improve the clarity in the final version of the paper.
>
> (3) Relationships between technical equations and clustering phase are sparse:
>
> About the technical equations and clustering phase mentioned by the reviewer, the Eq.2-3-4 are the dynamical equations for self-organizing updating process. Once the dynamic (i.e., SyncMap space) is well-organized, we can perform any algorithms suitable for spatial clustering to “read out” the chunks. Thus, the clustering phase is not the key part of our algorithm. Besides, we specify the parameters for DBSCAN in the experiment section.
>
> (4) Stochastic block model and Markov model:
>
> Previously, we were not aware of methods of SBM for model-based clustering of graphs. After a more general search, we obtained many works mixing these concepts, and some tried to solved problems similar to ours. For example, a paper on AAAI “On the Scalable Learning of Stochastic Blockmodel”, use several SBM models to learn community including Karate and Dolphins networks. These SBM algorithms includes BLOS, GSMDL VBMOD SICL and SILvb. Below is the NMI comparison of those model with our algorithms.
>
> == Karate Network (2 classes)
>
> ====BLOS: 0.83
>
> ====GSMDL: 0.75
>
> ====VBMOD: 0.83
>
> ====SICL: 0.79
>
> ====SILvb: 0.77
>
> ====Ours: 0.76
>
> == Dolphins Network (2 classes)
>
> ====BLOS: 0.66
>
> ====GSMDL: 0.55
>
> ====VBMOD: 0.62
>
> ====SICL: 0.36
>
> ====SILvb: 0.38
>
> ====Ours: 0.86
>
> As shown in the results, Symmetrical SyncMap yields comparable result in Karate network, while having the highest NMI among all SBM models in Dolphins problem. We will add them as comparison for the final version of the paper. Here, we will incorporate these citations in our paper and rephrase some parts to give due credit to these researchers.
>
> In terms of the Markov model, we further found that if the problem definition is expanded so that each state is associated with a noisy observation (say a multivariate Gaussian with a state-dependent mean and covariance) and the state itself is unobserved, then the range of applications will be much more vast. In this case, the problem is kind of like fitting a Hidden Markov Model but rather than explicitly finding the transition matrix, finding the repeating sequences of hidden discrete states from the noisy observations. A recent (potentially) relevant paper in this regard is "Variational Predictive Routing with Nested Subjective Timescales" by Zakharov et al. ((https://arxiv.org/pdf/2110.11236.pdf); also see the section "Hierarchical temporal structure" in their Related Work for more examples). We will also add them as comparison for the final version of the paper.

---

### Official Review · Reviewer_C5fg · 2022-10-28

**Confidence:** 2
**Correctness:** 3
**Technical Novelty And Significance:** 3
**Empirical Novelty And Significance:** 3
**Recommendation:** 5

**Clarity, Quality, Novelty And Reproducibility:**

Clarity is the main problem of the paper. The description of SyncMap is hard to follow. In the figures, we see $x_t$, but I can't quite follow how it corresponds to $s_t$.

The paper mainly builds upon SyncMap, so it's not entirely novel, but the modifications it proposes are well-motivated and interesting.

**Strength And Weaknesses:**

The intuition of chunking and learning repetitive patterns and generalization as a graph clustering problem makes it easy to grasp the general thrust of the paper. The idea of balancing the positive and negative sets is well-motivated. The experimental results are strong, too.

The paper hurts the most on clarity. I found the description of SyncMap very hard to follow. In particular, it's still not clear to me how states transition and how $t_a$ is determined. May it would be helpful to explicitly walk through an iteration with a concrete example.

Real-world experiments seems to be on datasets that are very small with few chunks. I would have expected to see something NLP-related  since Word2Vec and linguistics is mentioned.

**Summary Of The Paper:**

Symmetrical SyncMap builds upon SyncMap as an unsupervised clustering algorithm on graphs. To evaluate the behavior, chunking was formalized into CGCP (Continual General Chunking Problem) and becomes a graph clustering problem. Each node in the graph is assigned a weight vector. A key aspect of updating the weight vector are set the set of positive nodes and negative nodes. In the original SyncMap, these would become unbalanced. Symmetrical SyncMap proposals a sampling scheme so that the number of nodes in each set is equal. The method is then evaluated on both synthetic and real world data with very good results.

**Summary Of The Review:**

The paper in its current state is hard to follow, but it does seem to have potential to be interesting. I think increased clarity and experiments on larger more interesting datasets would help a lot.

---

> ### Author Response · Authors · 2022-11-14
> **Thank you for your review**
>
> Thank you for a thorough review and questions that will help us improve the manuscript. Please see below for answers to your questions.
>
> (1) Input encoding lacks clarity:
> In the encoding scheme, we introduce a raw sequence $S=\{s_1, s_2,…s_t,…\}$, and an encoded sequence $X=\{x_1, x_2,…x_t,…\}$. Specifically, at each time step $t$, $s_t$ is like a one-hot state vector in which only one state is activated (i.e., “current state” = True, while other states = False). $t_a$ is updated, if state transition happens (i.e., a new state is activated different from the previous state), and this particular state will keep activating within the time-delay period (i.e., $tstep$). After $tstep$, it will be a state transition to another state, and new $t_a$ will be determined.
>
> By encoding $S$ to $X$, the influence of the state activation in a particular time step, whose value is exponentially decaying, would last for “m*tstep” steps. We then set a threshold to determine whether some states belong to “recent activated state” and put them into Positive Set (PS).
> We thank the reviewer for pointing out the lack of clarity, and we will define and illustrate the input encoding more clearly in the final version of the paper.
>
> (2) Larger dataset related to NLP:
> We agree with the reviewer that larger real-world problems such as NLP problems need to be investigated. In this paper, we walk towards solving real-world problems by incorporating imbalance in dynamical problems and analyzing how it affects the previous working learning system, i.e., SyncMap. We will increasingly walk towards applications as we did in this paper, yet our applications will be in problems with a clear ground truth or well-accepted evaluation metric. Nevertheless, we conducted further experiments with CGCP problems with large-scale problem shown below.
>
> ==Imbalanced CGCP with 300-dimensional input (50x4 + 20x2 + 10x4 + 5x4)
>
> ==== Four chunks with 50 states + Two chunks with 20 states +Four chunks with 10 states + Four chunks with 5 states
>
> == NMI score
>
> ====Symmetrical SyncMap (k=3): 0.31±0.03
>
> ====Symmetrical SyncMap (k=15): 0.93±0.03
>
> ====Original SyncMap (k=3): 0.34±0.02
>
> ====Original SyncMap (k=15): 0.72±0.08
>
> ====Modularity Max: 0.60±0.0
>
> ====Word2vec: 0.78±0.16
>
> It is observed that the proposed Symmetrical SyncMap can achieve highest NMI score by increasing the dimension of the dynamics, allowing more movements of the weight nodes. Meanwhile, higher dimension problems up to 4000-D are in our another ongoing work with image processing using SyncMap.
>
> To sum up, in order to tackle multiple types of real-world problems, we are focusing on features of real-world problems, adding one at a time. Before walking towards larger real-world problems, we need to clarify the challenges of every feature present in them and how they can be tackled.

---

### Decision · Program_Chairs · 2023-01-20

**Decision:**

Reject

**Justification For Why Not Higher Score:**

As it stands none of the reviewers were willing to champion this paper for acceptance and it seems like this paper needs some further work to be ready for publications both in terms of experiments and also improving the writing.

**Justification For Why Not Lower Score:**

N/A

**Metareview: Summary, Strengths And Weaknesses:**

### Summary

This paper builds up on the unsupervised chunking algorithm called SyncMap and develops symmetrical SyncMap algorithm. The proposed method can be formalized as CGCP (Continual General Chunking Problem) and that makes it a a graph clustering problem. The nodes in the graph are represented as weight vectors and they are balanced by the number of updates from positive and negative feedback loops. The paper demonstrated the effectiveness of the proposed method on multiple synthetic and real-world datasets.

Below I will point out to some of the pros and and cons of the paper as indicated by the reviewers which guided the final decision for this paper:

### Strengths
- The idea of balancing the positive and negative sets is well-motivated and this is an interesting method.
- The experimental results are convincing.
- The paper studies an interesting problem.

### Weaknesses
- The paper hurts the most on clarity,  the description of SyncMap is very hard to follow. Writing needs improvements.
- Real-world experiments seems to be on datasets that are very small with few chunks. No NLP experiments.
- The technical contribution seems incremental.
- More ablations regarding to symmetrical activation and memory window would be helpful.


### Decision
This paper studies an interesting problem and proposes an approach that seems to work well in the settings where they have tested. However, as it stands none of the reviewers were willing to champion this paper for acceptance and it seems like this paper needs some further work to be ready for publications both in terms of experiments and also improving the writing.